# The Application of the Fuzzy Analytic Hierarchy Process in the Assessment and Improvement of the Human Settlement Environment

**Yangli Zhang** [1,*] and **Qiang Fan** [2]

1   College of Marxism, Sichuan University, Chengdu 610065, China
2   State Key Laboratory of Hydraulics and Mountain River Engineering, Sichuan University, Chengdu 610065, China; scufanqiang@stu.scu.edu.cn
*   Correspondence: zhangxiaoli@stu.scu.edu.cn; Tel.: +86-028-8599-6691

**Abstract:** With the development of urbanization in developing countries, some large cities have experienced rapid population growth and industrial expansion in a short period of time. In order to reasonably expand the scale of the city and guide the orderly outward movement of population and industries, it is urgent to improve the human settlement environment (THSE) in the surrounding areas of large cities. In the case of limited financial funds, different areas around the city need to be improved one by one according to the order of improvement grades. Since THSE is a comprehensive system involving multiple levels and indexes, it is difficult to assess it in a simple way. The previous assessment of THSE mainly focused on qualitative and semi-quantitative aspects, with poor accuracy. In this paper, the author takes JianYang County under the jurisdiction of Chengdu City in Southwest China as an example and uses the Fuzzy Analytic Hierarchy Process (FAHP) to quantitatively calculate the improvement grade of THSE in 55 townships of JianYang County. The author carried out an investigation for more than one year. According to the actual situation of JianYang County, five primary indexes and 22 secondary indexes were selected to establish a comprehensive evaluation index system. This index system contains 1210 statistical data points, and more than 30,000 data points were calculated and derived in this article. Finally, the author calculated the improvement grade of 55 townships by FAHP quantitatively and carried out a horizontal comparison of townships within the same grade to further determine the order of improvement of THSE.

**Keywords:** the human settlement environment (THSE); Fuzzy Analytic Hierarchy Process (FAHP); quantitative calculation; improvement grade; JianYang County

## 1. Introduction

The concept of the human settlement environment (THSE) began with the theory of "human settlement" proposed by Doxiadis C A in 1975 [1]. The Chinese scholar Liangyong Wu first advocated the establishment of THSE science in 1990 [2]. He believes that THSE is a place where human beings live together, which is composed of the natural system, human system, social system, living system, and environmental system. With the continuous progress of human civilization, THSE is also constantly improved. It can be said that the history of human development is also a history of the improvement of THSE. In the contemporary world, THSE has become an important indicator for measuring the living standard of a country and a region [3]. Although the concept of THSE was put forward earlier, the research on the Ekistics theory and THSE remain hot topics worldwide. Regarding solving the problem of THSE, Western developed countries are at the forefront of the world, but the developing countries with large populations and poor areas still facing huge problems regarding THSE. Moreover,

the assessment method of THSE has been changing dynamically as time progresses. As the largest developing country in the world, the process of urbanization in China has been accelerating since the reform and opening up. Some super-large cities are suffering from "Big City Diseases" due to rapid population growth and industrial expansion. Excessive population and industries have gathered in the main urban area of the cities, resulting in housing shortages, traffic congestion, air pollution, natural environment deterioration, and other problems, which seriously affect the quality of life of urban residents [4]. To solve the "Big City Diseases", local governments need not only to improve the living conditions of residents in the main urban areas but also, more importantly, the need to expand the city scale reasonably, making great efforts to improve THSE of the surrounding areas and guiding the orderly outward movement of population and industries. The development of mega-cities in major, Western, developed countries has basically gone through this process [5–7].

Before starting to improve THSE in the surrounding areas of the city, it is necessary to assess the current situation of THSE in the selected areas, and priority should be given to areas where improvement is urgently needed. Therefore, to reasonably classify THSE in different areas is very important. THSE is a multi-level and multi-type spatial system, involving many aspects such as politics, economy, culture, society, population, resources and the natural environment, which is very complicated when carrying out an assessment [8–10]. Many experts and scholars have carried out research into the assessment and improvement of the human settlement environment (THSE).

Daisy Das found that dimensions of quality of life in Guwahati are closely linked with the economic, social and physical environment. To improve the quality of life, a joint evaluation mechanism must be established [11]. According to the survey data, Lazauskaitė, D. et al. constructed a group of comprehensive evaluation systems of rural residential environment quality indexes based on subjective evaluation [12]. Yi Wang et al. took Zhejiang province, China, as an example and constructed the assessment framework of human habitability from three aspects—the ecological environment, economic development, and public service—and evaluated human habitability based on questionnaire survey [13]. Bonaiuto M. et al. introduced some related tools of UN-Habitat CPI and discussed the relationship between the overall living environment, community ownership and overall satisfaction of residents in Tabriz, Iran [14]. Based on the entropy analysis method and the ArcGIS spatial analysis method, Zhanhua Jia et al. explored the spatial-temporal characteristics of the livability levels of 37 cities in Northeast China from 2007 to 2014 [15]. Yan M et al. studied the environmental quality of urban residential areas by using the image data collected by a high-resolution remote sensing satellite in China [16]. Huiping H. et al. used the remote sensing data collected by the Gaofen 2 satellite and remote sensing data collected by some enterprises to establish an adaptive evaluation system for residential land and construction land in Haidian District, Beijing, and performed relevant semi-quantitative calculations [17].

Ning T. et al. used an entropy method to calculate the quality and spatial distribution of the rural human settlement environment in 37 counties of Chongqing [18]. Based on the SBM-DDF model, Ning C. et al. discussed green development and new urbanization in China [19]. Considering the International and Russian experience of the utilization of solid waste, P. A. Lavrinenko analyzed the characteristics of investment projects that are aimed at solving environmental problems [20]. Nany Yuliastutil et al. proposed how local governments and communities should make decisions when building ecological villages with a good living environment [21]. Chiang, C.-L. et al. used the Analytic Hierarchy Process (AHP) method and three indicators to evaluate livable urban environments and calculated each indicator's weight [22]. Liaghat, M. explored and selected some evaluation indexes of coastal tourist destinations in the Dixon port area of Malaysia by using AHP alone [23]. Xia Z. et al. established the theoretical framework of rural human settlements quality assessment, including hardware facilities and soft power in rural areas. In addition, six provinces in Central and Eastern China were investigated and examined by the structural equation model [24]. E. Shcherbin et al. studied several factors influencing the development of the rural settlement environment and compiled a settlement potential cartogram for the Mogilev region in Russia [25]. Urban Benchmarking is a

tool to scan the potentials of European cities. It is meant as a practical tool for comparing cities and city-regions based on pan-European territorial evidence produced by ESPON [26,27]. The ESPON 2013 Programme, the European Observation Network for Territorial Development and Cohesion, was adopted by the European Commission on 7 November 2007. The Urban Benchmarking Tool and the CityBench Tool are widely used in Europe [28,29]. Some other experts and scholars have also carried out some relevant research works [30–44].

On the whole, experts and scholars place different emphases on the assessment methods of THSE, but there are also the following shortcomings:

1.  Most of these studies focused on qualitative aspects, and rare quantitative calculations were made in the assessment of THSE.
2.  Some scholars used the principal component analysis method, the grey relational analysis method, and the SBM-DF model to evaluate THSE. These semi-quantitative methods are not perfect in theory, and the calculation process is inconsistent with human logic judgment thinking.
3.  Some scholars used the entropy weight TOPSIS method and Delphi method, meaning that the calculation process was too complicated to give a clear assessment result of THSE.
4.  The Urban Benchmarking Tool and the CityBench Tool are only targeted at the European regions, and there are no data from other regions. The evaluation methods and evaluation indexes of these two tools cannot solve the problem of the human settlement assessment.

In view of this, based on the analysis of previous research results, this paper adopts the Fuzzy Analytic Hierarchy Process (FAHP), which is more perfect in theory, clearer in level and simpler in its calculation process to evaluate THSE [45]. FAHP has obvious advantages in solving fuzzy and unstructured decision-making problems [46,47]. This method can make the subjective judgment process mathematical and logical, and the judgment process is more consistent with human thinking, which is very suitable for solving the problem of human settlement environment (HSE) assessment. In this paper, the author takes JianYang County under the jurisdiction of Chengdu City in Southwest China as an example and uses the FAHP to quantitatively calculate the improvement grade of THSE in 55 townships of JianYang County.

## 2. Materials and Methods

### 2.1. Study Area

Chengdu is the capital city of Sichuan Province in China. It is also the most important central city and mega-city in Southwest China. The development of Chengdu for more than 3000 years has taken place in the plain area between the Longmen Mountains and the Longquan Mountains (as shown in Figure 2), expanding circle by circle in the way of single-center gathering. Since the reform and opening up, with the increasing level of economic development in Chengdu, the industrial scale and population-scale have experienced explosive growth. At present, Chengdu exhibits the problems of housing difficulties, traffic congestion, air pollution, and environmental degradation. The original urban spatial structure can no longer satisfy the needs of the modernization development of the mega-city. Therefore, it is particularly urgent to expand the city scale to the surrounding areas. As Chengdu is located at the northwest edge of the Sichuan Basin, the special geographical structure determines that this city can only expand to the east or south. Therefore, the Chengdu Municipal Government proposed "Chengdu's overall plan for implementing the strategy of 'Eastward' (2018-2035)" in 2018 (File S1) and plans to build a new eastern city by crossing the Longquan mountains in nearly 20 years. While solving Chengdu's "Big City Disease", this can achieve the purpose of improving the human settlement environment (THSE) and reshaping the spatial structure of urban areas [48].

The main implementation area of the new eastern city is located in JianYang County, the geographical location maps are shown in Figures 1 and 2. JianYang County has a total area of

2213.5 km², a population of 1.5 million and 55 townships and is currently managed by Chengdu city. The residents in the county are mainly engaged in agricultural product planting, breeding and manufacturing, and the economic development level of each township is quite different [49,50]. In order to better conduct the population transfer and build a livable urban environment, according to the requirements of the "Plan", the Chengdu Municipal Government plans to carry out infrastructure improvement and THSE improvement in JianYang County in batches and regions over a period of 20 years. Therefore, it is particularly important to clarify the characteristics of THSE in JianYang County and to formulate a plan for the treatment of different human settlement environments according to local conditions. As a large number of financial resources are needed to build the new eastern city, it is impossible to invest huge financial funds in a short period of time. The Chengdu Municipal Government can only pay part of the annual expenses for the improvement of THSE in JianYang County. In order to achieve the maximum effect of improvement with limited investment, a reasonable improvement plan must be formulated. Therefore, it is necessary to assess THSE of the 55 townships in JianYang County and classify the urgency degree of improvement of the 55 townships. According to the improvement priority, priority should be given to the areas that need to be improved urgently, and the areas that do not need urgent improvement can be treated gradually in the later stage.

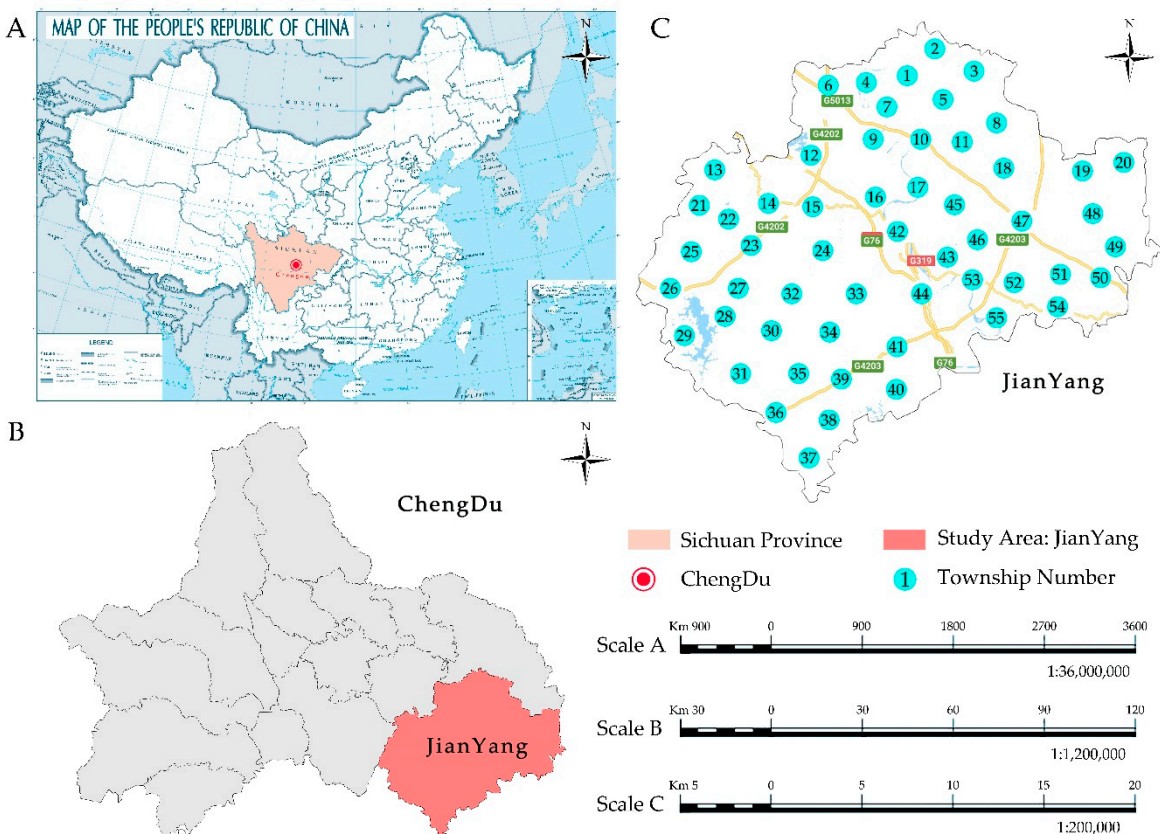

**Figure 1.** The location of the study area. (**A**) Location of Chengdu city in China. (**B**) Location of JianYang County of Chengdu city. (**C**) Administrative division of the study area: JiangYang County. (The maps were created in ArcGIS software Version 10.2. A high-resolution image can be found in Supplementary Materials).

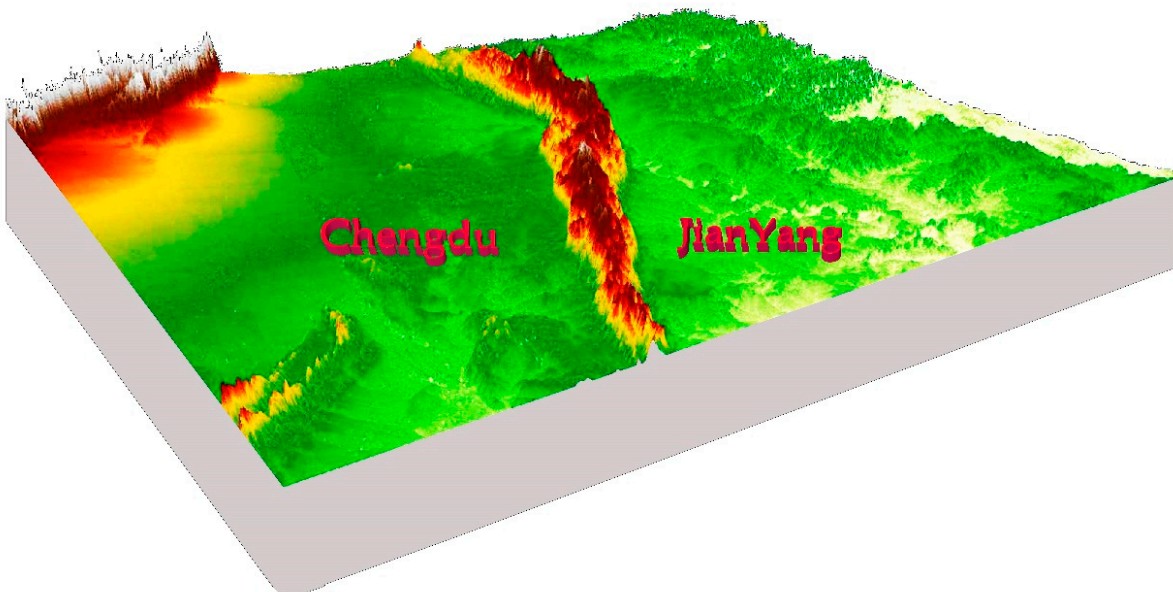

**Figure 2.** The geographical location map (3D Space Diagram, A high-resolution image can be found in Supplementary Materials).

*2.2. Research Method*

In carrying out research on the human settlement environmental (THSE) assessment and improvement, some other qualitative and semi-quantitative methods have also been adopted by some experts and scholars. These methods mainly include the principal component analysis method, entropy weight TOPSIS method, grey relational analysis method, Delphi method, SBM-DDF model, and so on. However, compared with the FAHP, these methods generally have the disadvantages of lacking rigorous theory, strong subjective judgment consciousness, poor calculation accuracy and so on. FAHP has the advantages of a complete theoretical system, a concise calculation process, and calculation results in accordance with human thinking logic, etc. It is suitable for solving problems that are difficult to quantify in the process of THSE assessment.

The Fuzzy Analytic Hierarchy Process (FAHP) is a kind of calculation method combining the Analytic Hierarchy Process (AHP) and fuzzy comprehensive evaluation method (FCEM). AHP is a decision-making method that decomposes the decision-making elements into objectives, criteria, schemes and other levels, on which qualitative and quantitative analysis are carried out. This method was first proposed by T.L. Saaty, a professor at the University of Pittsburgh [51,52]. The characteristic of this method is to make the decision process mathematical by using less quantitative information. The basic idea of AHP is to decompose the problem itself in layers, forming a hierarchical structure from the bottom to top, which is used to solve unstructured decision problems [53]. Among many evaluation methods, AHP has a complete theory and rigorous structure, which can make the subjective judgment process mathematical and thoughtful. Its decision-making basis is easy to be accepted by people, and it is more suitable for solving complex social science evaluation problems.

However, AHP also has a limitation, that is, when there are more evaluation indexes, the consistency of its thinking is hard to guarantee. In order to solve this problem, the FCEM based on fuzzy mathematics was applied to AHP in this article, which can overcome the shortcoming of AHP. The core of the FCEM is to change qualitative subjective judgment into quantitative objective judgment based on membership theory, that is, a comprehensive, accurate and objective evaluation of things or objects restricted by many factors was made according to fuzzy mathematics [54,55].

The problem-solving process of FAHP can be divided into four basic steps, as follows: (1) select the appropriate evaluation indexes based on the characteristics of the evaluation objects, and establish the comprehensive evaluation index system based on the evaluation indexes, (2) use AHP to calculate the

weight $w_i$ (*i* represents the number of evaluation indexes) of the evaluation index in the corresponding index level, and determine the weight vector **W** of the comprehensive evaluation index system, **W**= $(w_1, w_2, \ldots w_i)$, (3) use FCEM to calculate the evaluation vector $\mathbf{e}_t$=$(e_1, e_2, \ldots e_j)$ (*t* represents the number of evaluation objects, *j* represents the number of evaluation grade) of each evaluation index in turn, and determine the evaluation matrix **E** of all evaluation indexes, **E**=$(e_1, e_2, \ldots e_t)^T$, and (4) calculate the comprehensive evaluation set **C** of each evaluation object based on the theory of fuzzy mathematics, **C**=$(c_1, c_2, \ldots c_j)$, where the maximum value of $c_j$ corresponds to the evaluation grade of the research object.

### 2.3. Data Collection and Analysis

The author has been engaged in the research into sustainable urban and rural development and the human settlement environment (THSE) for a long time. The team to which the author belongs was invited to participate in the preliminary work of human settlement environment assessment and improvement in JianYang County, Chengdu. The author has made a detailed investigation of the basic situation of JianYang County for more than one year. On the basis of consulting a large amount of official data and on-the-spot investigation visits, the author has obtained more detailed data of each township.

In order to truly reflect the actual situation of THSE in JianYang County, the author takes the comprehensiveness and accuracy of data investigation as the most important principle when collecting and analyzing the statistics. THSE of a region is synergistic with multiple factors. Therefore, the author has fully considered the actual development condition of JianYang County during the establishment of a comprehensive evaluation index system. In the process of selecting evaluation indexes, they include not only the macro level and micro level, but also quantitative and quality factors. Finally, the author took 55 townships in JianYang County as the research object and selected five representative primary indexes and 22 secondary indexes to study the local THSE. The five primary indexes are economic conditions, living conditions, traffic conditions, public services, and the ecological environment. The comprehensive evaluation index system is shown in Table 1. Detailed data of each township are shown in Table 2.

**Table 1.** The comprehensive evaluation index system.

| Primary Index | Symbol | Secondary Index | Symbol | Unit |
|---|---|---|---|---|
| Economic Conditions | F | Per capita local GDP | $F_1$ | USA $ |
| | | Per capita disposable income of residents | $F_2$ | USA $ |
| | | Family average Engel coefficient | $F_3$ | % |
| | | Proportion of added value of tertiary industry in local GDP | $F_4$ | % |
| | | Proportion of Municipal Public facilities Construction Capital Investment in local GDP | $F_5$ | % |
| Living Conditions | L | Harmless treatment rate of domestic garbage | $L_1$ | % |
| | | Sewage treatment rate | $L_2$ | % |
| | | Gas penetration rate | $L_3$ | % |
| | | Tap water penetration rate | $L_4$ | % |
| | | Per capita housing area | $L_5$ | $m^2$ |
| Traffic Conditions | T | Road hardening rate | $T_1$ | % |
| | | Highway mileage per 100 people | $T_2$ | m |
| | | Ordinary road mileage per 100 population | $T_3$ | m |
| | | Number of buses per 1,000 people | $T_4$ | |
| Public Services | P | Number of clinics or hospitals per 1000 people | $P_1$ | |
| | | Number of schools per 1000 people | $P_2$ | |
| | | Stadium area per 1000 people | $P_3$ | $m^2$ |
| | | Number of urban comprehensive fire control facilities per 1000 people | $P_4$ | |
| | | Growth rate of public security cost | $P_5$ | % |
| Ecological Environment | E | Green coverage rate of built-up area | $E_1$ | % |
| | | Proportion of days with good air quality throughout the year | $E_2$ | % |
| | | Per capita park green area | $E_3$ | $m^2$ |

**Table 2.** Detailed data of each township in JianYang County

| Evaluation Index | | F | | | | | L | | | | | T | | | | P | | | | | E | | |
|---|---|---|---|---|---|---|---|---|---|---|---|---|---|---|---|---|---|---|---|---|---|---|---|
| Code Number | Place Name (Township) | $F_1$ USA\$ | $F_2$ USA\$ | $F_3$ % | $F_4$ % | $F_5$ % | $L_1$ % | $L_2$ % | $L_3$ % | $L_4$ % | $L_5$ m² | $T_1$ % | $T_2$ m | $T_3$ m | $T_4$ | $P_1$ | $P_2$ | $P_3$ m | $P_4$ | $P_5$ % | $E_1$ % | $E_2$ % | $E_3$ m² |
| 1 | Hongyuan | 3,000.61 | 1,621.76 | 43.7 | 20.1 | 35.4 | 45.6 | 24.5 | 45.3 | 64.3 | 48.6 | 73.7 | 0.0 | 50.6 | 2.4 | 1.2 | 0.4 | 17.4 | 4.2 | 5.4 | 62.5 | 82.2 | 3.5 |
| 2 | Tonghe | 2,966.73 | 1,512.32 | 43.8 | 20.4 | 33.3 | 46.4 | 25.1 | 46.2 | 65.6 | 47.5 | 63.4 | 0.0 | 60.3 | 3.2 | 2.3 | 0.6 | 19.3 | 5.3 | 6.3 | 53.4 | 81.6 | 3.1 |
| 3 | Xinxing | 1,922.09 | 1,257.16 | 44.9 | 21.2 | 36.7 | 35.8 | 20.4 | 46.3 | 66.2 | 46.3 | 62.4 | 0.0 | 58.3 | 3.3 | 2.1 | 0.6 | 16.5 | 5.7 | 5.8 | 61.7 | 81.4 | 4.3 |
| 4 | Lingxian | 3,616.32 | 1,757.17 | 43.2 | 22.3 | 32.1 | 50.3 | 35.5 | 47.5 | 60.3 | 48.8 | 78.5 | 5.3 | 67.7 | 3.1 | 1.8 | 0.7 | 17.2 | 5.4 | 7.6 | 72.6 | 80.5 | 4.7 |
| 5 | Sanxing | 2,912.93 | 1,409.83 | 44.5 | 20.5 | 29.5 | 47.8 | 33.4 | 42.3 | 59.4 | 45.6 | 65.3 | 0.0 | 62.1 | 2.5 | 2.3 | 0.5 | 14.5 | 6.5 | 8.5 | 67.8 | 80.8 | 5.2 |
| 6 | Zhoujia | 3,027.67 | 1,542.97 | 43.7 | 20.8 | 34.6 | 49.2 | 34.7 | 48.4 | 62.3 | 42.1 | 67.4 | 18.2 | 54.2 | 1.9 | 2.1 | 0.3 | 20.4 | 4.2 | 6.2 | 66.9 | 84.9 | 3.2 |
| 7 | Zhuangxi | 3,269.72 | 1,925.75 | 43.1 | 30.4 | 40.3 | 51.6 | 36.3 | 49.5 | 63.5 | 39.5 | 73.5 | 7.5 | 55.6 | 2.6 | 2.4 | 0.5 | 23.1 | 5.3 | 7.5 | 71.3 | 78.1 | 3.5 |
| 8 | Tashui | 2,369.01 | 1,282.95 | 45.0 | 19.2 | 30.1 | 36.2 | 22.9 | 42.2 | 57.3 | 40.3 | 71.3 | 0.0 | 57.3 | 2.3 | 2.2 | 0.4 | 18.5 | 4.7 | 6.8 | 68.5 | 87.7 | 4.2 |
| 9 | Yangma | 4,810.92 | 2,354.16 | 42.6 | 33.6 | 44.3 | 53.2 | 36.8 | 49.5 | 66.5 | 38.7 | 84.5 | 19.4 | 67.8 | 1.9 | 1.5 | 0.3 | 16.9 | 3.8 | 7.1 | 76.4 | 86.3 | 2.7 |
| 10 | Pingwo | 2,589.29 | 1,389.13 | 44.6 | 18.5 | 28.6 | 46.1 | 33.1 | 39.6 | 61.8 | 37.8 | 64.7 | 31.3 | 66.9 | 3.4 | 3.2 | 0.5 | 20.7 | 5.6 | 5.3 | 68.2 | 86.8 | 3.3 |
| 11 | Qinglong | 3,318.40 | 2,081.84 | 42.4 | 35.2 | 32.4 | 53.5 | 37.3 | 40.7 | 67.9 | 41.2 | 68.9 | 10.2 | 62.4 | 3.1 | 3.3 | 0.6 | 20.3 | 6.8 | 5.7 | 78.3 | 83.6 | 3.5 |
| 12 | Shipan | 3,036.77 | 1,400.00 | 44.1 | 27.5 | 30.4 | 47.3 | 33.6 | 39.8 | 57.3 | 38.6 | 66.7 | 16.1 | 65.1 | 1.5 | 1.6 | 0.3 | 15.6 | 3.9 | 6.6 | 56.8 | 82.2 | 2.8 |
| 13 | Laojunjin | 2,977.79 | 1,457.21 | 43.9 | 24.5 | 27.4 | 37.5 | 24.6 | 41.4 | 60.3 | 39.1 | 65.4 | 7.8 | 70.3 | 3.5 | 2.5 | 0.4 | 19.3 | 5.2 | 7.9 | 59.3 | 78.6 | 2.9 |
| 14 | Jiajia | 5,260.68 | 2,779.62 | 42.5 | 42.6 | 41.5 | 64.2 | 38.2 | 52.5 | 70.2 | 38.5 | 79.2 | 17.8 | 50.8 | 1.8 | 1.5 | 0.3 | 17.2 | 3.7 | 5.9 | 62.1 | 77.5 | 2.5 |
| 15 | Taipingqiao | 2,617.01 | 1,540.47 | 43.8 | 30.5 | 42.3 | 48.3 | 35.1 | 53.4 | 61.6 | 42.4 | 73.1 | 18.3 | 56.8 | 1.6 | 1.7 | 0.2 | 19.5 | 4.2 | 8.4 | 63.2 | 77.0 | 3.2 |
| 16 | Shiqiao | 5,319.41 | 2,406.85 | 41.5 | 40.2 | 46.3 | 56.3 | 39.3 | 57.3 | 74.8 | 35.6 | 80.2 | 18.6 | 67.8 | 2.1 | 2.3 | 0.4 | 23.3 | 3.7 | 9.3 | 70.5 | 80.5 | 3.1 |
| 17 | Shizhong | 4,390.97 | 2,320.00 | 42.3 | 42.4 | 41.2 | 62.1 | 45.3 | 53.1 | 75.1 | 36.3 | 80.5 | 15.6 | 66.9 | 2.2 | 1.6 | 0.4 | 25.6 | 3.8 | 8.8 | 57.7 | 81.1 | 3.4 |
| 18 | Sanhe | 2,388.26 | 1,161.50 | 44.6 | 29.4 | 36.7 | 48.4 | 36.4 | 40.2 | 72.9 | 42.7 | 64.2 | 6.5 | 57.9 | 1.8 | 1.8 | 0.3 | 16.3 | 3.3 | 6.8 | 65.4 | 81.4 | 4.6 |
| 19 | Jinma | 3,669.13 | 1,917.60 | 42.8 | 34.6 | 32.6 | 52.2 | 34.9 | 45.6 | 73.4 | 41.6 | 68.6 | 0.0 | 75.9 | 3.2 | 3.1 | 0.4 | 15.8 | 5.3 | 8.5 | 71.2 | 83.6 | 4.4 |
| 20 | Wuhe | 3,361.56 | 1,714.42 | 42.9 | 32.7 | 36.1 | 54.3 | 38.2 | 41.3 | 67.8 | 39.7 | 66.3 | 0.0 | 54.7 | 3.2 | 3.2 | 0.5 | 20.4 | 6.7 | 7.9 | 68.5 | 85.2 | 4.1 |
| 21 | Wuzhi | 2,951.69 | 1,528.65 | 43.5 | 25.8 | 33.5 | 47.8 | 37.1 | 42.2 | 65.3 | 38.4 | 65.7 | 0.0 | 63.5 | 2.9 | 2.6 | 0.6 | 22.1 | 5.2 | 9.7 | 73.6 | 85.2 | 3.6 |
| 22 | Gaoming | 2,946.65 | 1,771.03 | 43.1 | 27.4 | 40.6 | 46.4 | 31.2 | 46.3 | 68.7 | 40.6 | 74.5 | 13.7 | 62.1 | 2.7 | 2.4 | 0.5 | 17.9 | 5.1 | 8.6 | 67.1 | 84.4 | 3.7 |
| 23 | Tanguan | 5,857.24 | 3,248.88 | 39.4 | 39.5 | 36.4 | 58.1 | 40.2 | 57.3 | 73.2 | 37.5 | 73.5 | 23.2 | 63.7 | 3.1 | 3.5 | 0.3 | 18.6 | 4.7 | 8.2 | 66.8 | 79.5 | 2.6 |
| 24 | Hailuo | 5,061.99 | 2,662.82 | 40.8 | 38.4 | 37.4 | 56.3 | 41.3 | 56.5 | 71.7 | 36.7 | 76.8 | 0.0 | 54.6 | 3.1 | 3.1 | 0.4 | 19.6 | 6.3 | 8.4 | 76.3 | 77.5 | 2.9 |
| 25 | Wumiao | 4,589.29 | 2,569.05 | 41.2 | 36.3 | 33.2 | 52.3 | 42.5 | 52.1 | 75.8 | 38.1 | 77.4 | 0.0 | 53.7 | 2.5 | 2.5 | 0.5 | 15.7 | 5.8 | 7.8 | 73.6 | 76.2 | 3.2 |
| 26 | Danjing | 4,666.67 | 2,282.66 | 42.1 | 35.5 | 43.2 | 60.4 | 45.6 | 52.4 | 76.3 | 37.2 | 72.1 | 5.8 | 72.7 | 3.2 | 3.4 | 0.5 | 21.5 | 6.2 | 7.5 | 56.8 | 75.3 | 2.8 |
| 27 | Yucheng | 4,969.26 | 2,684.76 | 40.7 | 38.9 | 40.5 | 62.5 | 45.8 | 53.2 | 79.4 | 36.3 | 78.5 | 8.4 | 53.4 | 1.6 | 2.5 | 0.3 | 23.4 | 3.7 | 7.7 | 64.3 | 74.8 | 3.4 |
| 28 | Sancha | 5,309.74 | 2,779.74 | 40.3 | 40.5 | 42.1 | 58.7 | 39.7 | 49.6 | 71.4 | 35.4 | 79.3 | 0.0 | 59.2 | 1.4 | 1.8 | 0.2 | 25.9 | 3.8 | 5.9 | 66.2 | 74.0 | 2.3 |
| 29 | Xinmin | 3,763.47 | 2,184.05 | 42.5 | 41.1 | 46.3 | 56.4 | 38.9 | 48.5 | 68.1 | 38.6 | 71.2 | 0.0 | 69.8 | 3.2 | 2.9 | 0.4 | 21.7 | 5.4 | 6.3 | 58.4 | 73.4 | 2.9 |
| 30 | Futian | 5,542.97 | 2,925.51 | 40.2 | 43.6 | 32.1 | 63.7 | 46.8 | 50.3 | 69.2 | 39.2 | 71.8 | 0.0 | 74.2 | 3.3 | 3.8 | 0.5 | 17.2 | 6.3 | 7.5 | 68.2 | 73.2 | 2.4 |
| 31 | Dongjiageng | 4,690.13 | 2,622.75 | 40.8 | 37.4 | 29.6 | 65.9 | 46.3 | 49.7 | 65.7 | 40.3 | 76.5 | 7.9 | 70.8 | 3.2 | 3.6 | 0.5 | 16.8 | 5.3 | 6.9 | 54.8 | 72.3 | 2.5 |
| 32 | Caochi | 3,992.69 | 2,321.92 | 42.4 | 36.4 | 35.1 | 61.6 | 47.1 | 48.9 | 66.9 | 40.5 | 77.9 | 0.0 | 52.1 | 1.3 | 1.7 | 0.2 | 19.3 | 3.4 | 10.3 | 53.9 | 71.5 | 2.6 |
| 33 | Shibandeng | 5,475.15 | 3,426.78 | 39.2 | 45.6 | 42.9 | 66.4 | 47.2 | 49.6 | 70.3 | 38.7 | 84.6 | 9.2 | 57.8 | 2.1 | 2.5 | 0.4 | 20.4 | 4.7 | 9.5 | 57.2 | 71.2 | 3.1 |
| 34 | Lujia | 3,152.06 | 1,797.61 | 42.3 | 33.4 | 37.4 | 56.3 | 38.6 | 48.5 | 71.5 | 40.7 | 68.8 | 6.9 | 48.9 | 2.3 | 2.4 | 0.5 | 22.5 | 3.6 | 9.9 | 63.4 | 81.9 | 3.6 |
| 35 | Qingfeng | 2,846.39 | 1,256.33 | 45.2 | 26.3 | 25.7 | 47.5 | 36.1 | 52.5 | 71.7 | 43.4 | 60.3 | 13.6 | 45.6 | 1.6 | 1.6 | 0.3 | 23.6 | 3.7 | 11.4 | 67.5 | 83.3 | 4.6 |
| 36 | Zhenjin | 3,670.74 | 1,724.79 | 42.4 | 30.5 | 27.4 | 43.6 | 35.8 | 46.5 | 74.5 | 41.2 | 65.4 | 17.8 | 73.1 | 3.8 | 3.2 | 0.6 | 27.4 | 6.7 | 10.5 | 58.7 | 83.8 | 4.3 |
| 37 | Laolong | 4,100.38 | 1,923.05 | 42.1 | 32.6 | 32.5 | 56.3 | 43.2 | 43.2 | 72.6 | 43.6 | 66.2 | 0.0 | 72.9 | 3.3 | 2.6 | 0.4 | 28.3 | 7.2 | 10.2 | 75.4 | 84.7 | 4.5 |
| 38 | Wangshui | 2,779.11 | 1,426.49 | 44.1 | 27.6 | 20.8 | 47.8 | 36.5 | 41.1 | 66.4 | 41.8 | 64.8 | 15.4 | 60.2 | 1.6 | 1.7 | 0.5 | 25.1 | 4.3 | 9.6 | 72.8 | 80.5 | 4.8 |
| 39 | Leijia | 3,317.75 | 1,655.66 | 43.2 | 24.3 | 26.4 | 35.9 | 26.4 | 40.6 | 64.7 | 40.6 | 68.9 | 13.6 | 58.4 | 3.2 | 2.6 | 0.5 | 23.5 | 5.3 | 9.8 | 71.9 | 80.5 | 3.7 |
| 40 | Yongning | 2,878.86 | 1,408.21 | 44.3 | 25.6 | 24.3 | 39.2 | 33.9 | 39.7 | 63.8 | 43.3 | 75.6 | 20.3 | 69.4 | 2.6 | 1.9 | 0.4 | 21.7 | 3.4 | 8.4 | 73.6 | 79.5 | 3.5 |

**Table 2.** *Cont.*

| Evaluation Index | | F | | | | | L | | | | | T | | | | P | | | | | E | | |
| Code Number | Place Name (Township) | $F_1$ USA$ | $F_2$ USA$ | $F_3$ % | $F_4$ % | $F_5$ % | $L_1$ % | $L_2$ % | $L_3$ % | $L_4$ % | $L_5$ m$^2$ | $T_1$ % | $T_2$ m | $T_3$ m | $T_4$ | $P_1$ | $P_2$ | $P_3$ m | $P_4$ | $P_5$ % | $E_1$ % | $E_2$ % | $E_3$ m$^2$ |
|---|---|---|---|---|---|---|---|---|---|---|---|---|---|---|---|---|---|---|---|---|---|---|---|
| 41 | Jiangyuan | 2,664.07 | 1,412.78 | 44.4 | 28.9 | 30.1 | 43.6 | 36.7 | 42.3 | 61.4 | 45.6 | 73.8 | 14.3 | 45.2 | 1.4 | 1.2 | 0.2 | 17.8 | 2.7 | 6.7 | 73.4 | 79.2 | 4.1 |
| 42 | Jiancheng | 8,158.11 | 5,210.31 | 32.3 | 52.3 | 46.5 | 75.6 | 53.4 | 64.3 | 83.4 | 32.3 | 93.5 | 10.2 | 42.3 | 2.4 | 1.1 | 0.2 | 19.3 | 4.6 | 11.3 | 50.2 | 78.4 | 1.5 |
| 43 | Dongxi | 5,989.57 | 2,905.04 | 39.5 | 46.4 | 43.7 | 70.3 | 50.1 | 59.5 | 78.8 | 35.4 | 90.4 | 11.3 | 63.2 | 2.1 | 1.3 | 0.2 | 16.8 | 3.2 | 10.5 | 53.7 | 77.5 | 1.9 |
| 44 | Xinshi | 4,732.71 | 2,710.49 | 40.6 | 43.7 | 41.2 | 68.5 | 49.7 | 58.4 | 77.8 | 36.8 | 88.6 | 13.2 | 65.4 | 1.7 | 1.2 | 0.2 | 17.8 | 3.5 | 9.7 | 62.4 | 78.1 | 1.4 |
| 45 | Pingwu | 2,201.87 | 1,065.18 | 45.6 | 22.6 | 34.1 | 53.2 | 43.5 | 42.1 | 74.3 | 41.2 | 83.4 | 17.8 | 70.5 | 3.3 | 3.2 | 0.4 | 15.4 | 6.3 | 6.8 | 68.5 | 76.7 | 2.6 |
| 46 | Puan | 2,159.54 | 1,030.46 | 45.2 | 21.8 | 33.5 | 46.8 | 37.4 | 40.5 | 73.7 | 43.8 | 76.5 | 19.4 | 71.3 | 3.7 | 2.7 | 0.4 | 19.7 | 4.7 | 7.9 | 69.3 | 80.3 | 3.5 |
| 47 | Hefeng | 1,953.80 | 891.40 | 46.2 | 19.5 | 34.3 | 43.5 | 35.2 | 41.6 | 66.9 | 44.5 | 73.2 | 20.3 | 62.3 | 2.5 | 1.8 | 0.3 | 17.3 | 4.2 | 10.2 | 73.8 | 81.4 | 3.7 |
| 48 | Yunlong | 1,660.25 | 903.60 | 46.3 | 18.6 | 33.7 | 42.8 | 24.3 | 42.7 | 63.2 | 39.6 | 87.5 | 0.0 | 61.5 | 2.4 | 2.1 | 0.5 | 20.4 | 5.8 | 9.3 | 72.4 | 81.9 | 4.2 |
| 49 | Yongquan | 2,790.90 | 1,382.65 | 44.2 | 26.7 | 28.3 | 56.8 | 32.7 | 41.3 | 61.7 | 43.3 | 83.4 | 26.7 | 75.6 | 4.3 | 3.4 | 0.5 | 23.5 | 6.8 | 9.6 | 74.3 | 80.8 | 3.8 |
| 50 | Pingxi | 1,811.88 | 1,133.47 | 45.1 | 25.6 | 29.5 | 51.2 | 31.9 | 42.6 | 67.5 | 35.4 | 81.2 | 30.5 | 68.4 | 2.6 | 2.4 | 0.4 | 22.8 | 5.8 | 10.6 | 67.1 | 82.5 | 4.1 |
| 51 | Anle | 1,600.13 | 895.46 | 46.5 | 22.1 | 28.9 | 37.9 | 27.5 | 40.9 | 63.7 | 34.8 | 76.4 | 24.5 | 72.3 | 3.5 | 3.4 | 0.6 | 21.4 | 7.3 | 7.5 | 77.2 | 86.3 | 2.7 |
| 52 | Wuxing | 1,948.84 | 1,117.34 | 45.4 | 23.4 | 25.3 | 39.4 | 34.6 | 39.9 | 74.3 | 38.2 | 73.2 | 23.1 | 65.1 | 2.8 | 3.1 | 0.5 | 24.3 | 6.4 | 8.4 | 74.6 | 86.6 | 3.1 |
| 53 | Pingquan | 1,768.72 | 982.76 | 46.1 | 22.7 | 30.2 | 32.3 | 28.5 | 43.2 | 75.8 | 40.5 | 69.6 | 22.8 | 67.3 | 3.6 | 2.8 | 0.3 | 19.1 | 5.2 | 9.1 | 71.9 | 87.4 | 2.3 |
| 54 | Shijia | 3,851.67 | 2,084.19 | 42.7 | 37.8 | 26.7 | 48.7 | 38.6 | 47.5 | 68.9 | 44.6 | 70.2 | 16.8 | 72.5 | 3.5 | 3.2 | 0.5 | 18.2 | 6.1 | 10.4 | 68.4 | 84.4 | 3.3 |
| 55 | Feilong | 2,071.72 | 1,227.49 | 44.8 | 28.7 | 22.4 | 53.6 | 35.3 | 40.7 | 65.2 | 45.7 | 70.6 | 13.2 | 77.9 | 3.1 | 3.2 | 0.6 | 20.5 | 5.8 | 9.5 | 70.2 | 84.7 | 4.1 |

Sources of the statistical data include (1) Data obtained from the field survey by the author and her team for more than one year. (2) "China Statistical Yearbook", "China Rural Statistical Yearbook", "China Civil Administration Statistical Yearbook", "Chinese Social Statistical Yearbook", "Chinese Cultural and Cultural Relics Statistical Yearbook", "Sichuan Statistical Yearbook 2019", "Sichuan Statistical Yearbook 2018", "Sichuan Statistical Yearbook 2017", "Evaluation standard for sustainable development of Chinese Academy of Sciences", "Sichuan Rural Statistical Yearbook 2018", "Sichuan Rural Statistical Yearbook 2017", "Sichuan Rural Statistical Yearbook 2016", "Sichuan Ecological Province Construction Planning Outline", "JianYang County Records". (3) Official data from some official websites. (4) Some micro survey data are from social survey reports provided by Sichuan University, Sichuan Normal University and Renmin University of China.

## 3. The Application of FAHP and Results

### 3.1. Weight of Evaluation Index

### 3.1.1. Judgment Matrix

After the establishment of the comprehensive evaluation index system, the judgment matrix of the primary index and the secondary index can be constructed by AHP, and the weight value of each index can be calculated according to the judgment matrix. The key to constructing the judgment matrix is to compare the indexes of the same level in the comprehensive evaluation index system according to the "1–9 scale method" by experienced experts and score the evaluation index with an integer between 1~9 and its reciprocal according to the importance degree of the evaluation index [56,57]. The numerical scale and the significance of the "1–9 scale method" are shown in Table 3.

**Table 3.** 1–9 scale method.

| Significance | Numeric Scale |
|---|---|
| Two evaluation indexes have the same importance | 1 |
| Comparison of two evaluation indexes: index $\mu$ is slightly more important than index $\nu$ | 3 |
| Comparison of two evaluation indexes: index $\mu$ is obviously more important than index $\nu$ | 5 |
| Comparison of two evaluation indexes: index $\mu$ is much more important than index $\nu$ | 7 |
| Comparison of two evaluation indexes: index $\mu$ is extremely more important than index $\nu$ | 9 |
| Median value of the above adjacent judgments | 2,4,6,8 |
| If the importance of index $\mu$ to index $\nu$ is $d$, then the importance of $\nu$ to $\mu$ is $1/d$ | Reciprocal |

Among the experts who judged and scored the evaluation indexes, some of them are professors on the author's team. In order to ensure an objective and comprehensive scoring process, some experts from other organizations were also invited by the author. The scores of the evaluation indexes are shown in Table 4, the colors in the table are used to distinguish different judgment matrices and do not represent any other meaning.

**Table 4.** Evaluation index scoring table.

| | F | L | T | P | E | | $F_1$ | $F_2$ | $F_3$ | $F_4$ | $F_5$ |
|---|---|---|---|---|---|---|---|---|---|---|---|
| F | 1 | 3 | 5 | 6 | 1/3 | $F_1$ | 1 | 1/7 | 1/5 | 1/3 | 1/4 |
| L | 1/3 | 1 | 3 | 4 | 1/4 | $F_2$ | 7 | 1 | 3 | 6 | 5 |
| T | 1/5 | 1/3 | 1 | 3 | 1/6 | $F_3$ | 5 | 1/3 | 1 | 4 | 3 |
| P | 1/6 | 1/4 | 1/3 | 1 | 1/7 | $F_4$ | 3 | 1/6 | 1/4 | 1 | 1/3 |
| E | 3 | 4 | 6 | 7 | 1 | $F_5$ | 4 | 1/5 | 1/3 | 3 | 1 |
| | $L_1$ | $L_2$ | $L_3$ | $L_4$ | $L_5$ | | $P_1$ | $P_2$ | $P_3$ | $P_4$ | $P_5$ |
| $L_1$ | 1 | 3 | 5 | 4 | 1/3 | $P_1$ | 1 | 1/3 | 5 | 4 | 3 |
| $L_2$ | 1/3 | 1 | 4 | 3 | 1/4 | $P_2$ | 3 | 1 | 6 | 5 | 4 |
| $L_3$ | 1/5 | 1/4 | 1 | 1/3 | 1/7 | $P_3$ | 1/5 | 1/6 | 1 | 1/3 | 1/4 |
| $L_4$ | 1/4 | 1/3 | 3 | 1 | 1/6 | $P_4$ | 1/4 | 1/5 | 3 | 1 | 1/3 |
| $L_5$ | 3 | 4 | 7 | 6 | 1 | $P_5$ | 1/3 | 1/4 | 4 | 3 | 1 |
| | $T_1$ | $T_2$ | $T_3$ | $T_4$ | | | $E_1$ | $E_2$ | $E_3$ | | |
| $T_1$ | 1 | 5 | 3 | 7 | | $E_1$ | 1 | 1/3 | 2 | | |
| $T_2$ | 1/5 | 1 | 1/3 | 4 | | $E_2$ | 3 | 1 | 5 | | |
| $T_3$ | 1/3 | 3 | 1 | 5 | | $E_3$ | 1/2 | 1/5 | 1 | | |
| $T_4$ | 1/7 | 1/4 | 1/5 | 1 | | | | | | | |

### 3.1.2. Hierarchical Single Ranking calculation and Consistency Check

The eigenvector corresponding to the largest eigenvalue $\lambda_{\max}$ of the judgment matrix is recorded as **W'**, and the element value $w'_n$ ($n$ represents the number of evaluation indexes at current index level)

in the vector **W**′ represents the ranking weight of the relative importance of the lower-level evaluation index to the upper-level evaluation index. This calculation process is called a hierarchical single ranking. The consistency check refers to checking the consistency of logical judgment, for example, if $\mu$ is more important than $\nu$, $\nu$ is more important than $\varepsilon$, then $\mu$ must be more important than $\varepsilon$, which is the consistency of logical judgment—otherwise, the judgment will be wrong. In this section, the author takes five evaluation indexes in the primary index and five evaluation indexes under the "economic conditions" in the secondary index as examples for the demonstration calculation. The calculation process of other indexes is the same.

1. Carry out the hierarchical single ranking calculation and consistency check for 5 evaluation indexes in the primary index:

According to Table 4, the expert's scoring results for the five evaluation indexes in the primary index are defined as the judgment matrix $\mathbf{J}_0$, that is,

$$\mathbf{J}_0 = \begin{pmatrix} 1 & 3 & 5 & 6 & 1/3 \\ 1/3 & 1 & 3 & 4 & 1/4 \\ 1/5 & 1/3 & 1 & 3 & 1/6 \\ 1/6 & 1/4 & 1/3 & 1 & 1/7 \\ 3 & 4 & 6 & 7 & 1 \end{pmatrix}$$

**First**, we calculate the element product $P_r$ of each row of the judgment matrix $\mathbf{J}_0$ and the $n$th root $T_r$ of $P_r$:

$$P_r = \prod_{c=1}^{n} j_{rc} \tag{1}$$

$$T_r = \sqrt[n]{P_r}; (r = 1, 2, \cdots, n) \tag{2}$$

where $r$ represents the row number of $\mathbf{J}_0$, $c$ represents the column number of $\mathbf{J}_0$, and $n$ represents the number of rows and columns of the matrix.

Then, we bring in the data and calculate as follows:

$P_1 = 30$, $P_2 = 1$, $P_3 = 0.0333$, $P_4 = 0.002$, $P_5 = 504$; $T_1 = 1.97$, $T_2 = 1$, $T_3 = 0.51$, $T_4 = 0.29$, $T_5 = 3.47$

**Next**, we calculate the largest eigenvalue $\lambda_{\max}$ of the judgment matrix $\mathbf{J}_0$, where $\mathbf{J}_0 \cdot \mathbf{W}′ = \lambda_{\max 0} \mathbf{W}′$.

$$\lambda_{\max} = \sum_{r=1}^{n} \frac{j_{r1}T_1 + j_{r2}T_2 + \cdots + j_{rn}T_n}{nT_i}; r = 1, 2, 3 \cdots, n \tag{3}$$

Therefore, $\lambda_{\max 0} = 5.28$

**Then**, we check the consistency. Since the value of $\mathbf{J}_0$ is determined by multiple experts based on their respective professional experience, in order to prevent logical errors and contradictions in $\mathbf{J}_0$, a consistency check of the judgment matrix is also required. The common test index is the consistency ratio (CR). When CR ≤ 0.1, the judgment matrix $\mathbf{J}_0$ meets the calculation requirement, when CR > 0.1, the judgment matrix $\mathbf{J}_0$ needs to be re-determined. The consistency ratio (CR) is a function of the consistency index (CI) and random index (RI), CR = CI/RI.

If

$$CI = (\lambda_{\max} - n)/(n - 1) \tag{4}$$

then RI can be determined by Table 5 [58].

**Table 5.** Table of the random index (RI).

| Order of the Matrix ($n$) | 1 | 2 | 3 | 4 | 5 | 6 | 7 | 8 | 9 |
|---|---|---|---|---|---|---|---|---|---|
| **Random Index** | 0 | 0 | 0.58 | 0.9 | 1.12 | 1.24 | 1.32 | 1.41 | 1.45 |

Then, we bring in the data and calculate as follows:

CI $= \frac{5.28-5}{5-1} = 0.07$; CR $= \frac{0.07}{1.12} = 0.0625 < 0.1$, the consistency of $\mathbf{J}_0$ meets the requirement.

**Finally**, we normalize the eigenvector $\mathbf{W}'$, and the weight value $w'_n$ of 5 evaluation indexes in the primary index can be calculated.

$$w'_r = T_r / \sum_{r=1}^{n} T_r \tag{5}$$

The weight value of the primary index relative to the target is

$w'_{01}=0.27$, $w'_{02}=0.14$, $w'_{03}=0.07$, $w'_{04}=0.04$, $w'_{05}=0.48$

Therefore, the weight vector of the primary index $\mathbf{W}_0 = (0.27, 0.14, 0.07, 0.04, 0.48)$.

2. We carry out hierarchical single ranking calculation and consistency check for the five evaluation indexes under the "economic conditions" in the secondary index:

In the same way:

$$\mathbf{J}_1 = \begin{pmatrix} 1 & 1/7 & 1/5 & 1/3 & 1/4 \\ 7 & 1 & 3 & 6 & 5 \\ 5 & 1/3 & 1 & 4 & 3 \\ 3 & 1/6 & 1/4 & 1 & 1/3 \\ 4 & 1/5 & 1/3 & 3 & 1 \end{pmatrix}$$

$P_1 = 0.0024$, $P_2 = 630$, $P_3 = 20$, $P_4 = 0.04$, $P_5 = 0.8$; $T_1 = 0.3$, $T_2 = 3.63$, $T_3 = 1.82$, $T_4 = 0.53$, $T_5 = 0.96$

$\lambda_{max1} = 5.29$

CR $= 0.0725/1.12 = 0.065 < 0.1$

$\xi_{11} = 0.04$, $\xi_{12} = 0.5$, $\xi_{13} = 0.25$, $\xi_{14} = 0.07$, $\xi_{15} = 0.13$

$\mathbf{W}_1 = (0.04, 0.5, 0.25, 0.07, 0.13)$.

### 3.1.3. Hierarchical Total Ranking Calculation and Consistency Check

In the comprehensive evaluation index architecture model, the weight of each layer relative to the target is calculated layer by layer. This process is called hierarchical total ranking. In the comprehensive evaluation index system, if the $h$th layer contains $t$ evaluation indexes ($Q_1, Q_2, \ldots Q_t$), the ($h$+1)th layer contains $m$ evaluation indexes ($S_1, S_2, \ldots S_m$), and the weight value of $Q_i$ ($I = 1,2 \ldots t$) relative to the target is $w'_i$, the weight value of $S_j$ ($j = 1,2 \ldots m$) relative to $h$th layer is $\xi_{ij}$, then the weight value of $S_j$ to the target is $\beta_{ij}$.

$$\beta_{ij} = \sum_{j=1}^{m} w'_i \xi_{ij} \tag{6}$$

The weight values of the five evaluation indexes under the "economic condition" relative to the target are

$\beta_{11} = 0.27 \times 0.04 = 0.01$, $\beta_{12} = 0.27 \times 0.5 = 0.13$, $\beta_{13} = 0.27 \times 0.25 = 0.06$, $\beta_{14} = 0.27 \times 0.07 = 0.02$, $\beta_{15} = 0.27 \times 0.13 = 0.03$.

Therefore, $\mathbf{W}_{01} = (0.01, 0.13, 0.06, 0.02, 0.03)$.

Similarly, the largest eigenvalue $\lambda_{max}$ and the weight vector $\mathbf{W}$ of other 17 secondary evaluation indexes are as follows:

$\lambda_{max2} = 5.27$, $\mathbf{W}_{02} = (0.26, 0.14, 0.04, 0.07, 0.49)$;

$\lambda_{max3} = 4.18$, $\mathbf{W}_{03} = (0.56, 0.13, 0.26, 0.05)$;

$\lambda_{max4} = 5.31$, $\mathbf{W}_{04} = (0.26, 0.47, 0.04, 0.08, 0.14)$;

$\lambda_{max5} = 3.00$, $\mathbf{W}_{05} = (0.23, 0.65, 0.12)$.

The consistency check still needs to be carried out for hierarchical total ranking.

$$CR = \frac{\sum\limits_{i=1}^{t} w'_i CI_i}{\sum\limits_{i=1}^{t} w'_i RI_i} \tag{7}$$

Then, we bring in the data and calculate as follows:

$$CR = \frac{0.27 \times \frac{5.29-5}{5-1} + 0.14 \times \frac{5.27-5}{5-1} + 0.07 \times \frac{4.18-4}{4-1} + 0.04 \times \frac{5.31-5}{5-1} + 0.48 \times \frac{3-3}{3-1}}{0.27 \times 1.12 + 0.14 \times 1.12 + 0.07 \times 0.9 + 0.04 \times 1.12 + 0.48 \times 0.58} = 0.043 < 0.1$$

Therefore, the results of the hierarchical total ranking also have satisfactory consistency.

### 3.1.4. Final Weight Vector

This paper has determined the weight values of five primary evaluation indexes and 22 secondary evaluation indexes through hierarchical single ranking calculation, hierarchical total ranking calculation, and a consistency check. The results are shown in Table 6.

**Table 6.** Weight value calculation results.

| Primary Index | Weight Value | Secondary Index | Weight Value | Final Weight Value |
|---|---|---|---|---|
| F | 0.27 | $F_1$ | 0.04 | 0.01 |
|  |  | $F_2$ | 0.50 | 0.13 |
|  |  | $F_3$ | 0.25 | 0.06 |
|  |  | $F_4$ | 0.07 | 0.02 |
|  |  | $F_5$ | 0.13 | 0.03 |
| L | 0.14 | $L_1$ | 0.26 | 0.07 |
|  |  | $L_2$ | 0.14 | 0.04 |
|  |  | $L_3$ | 0.04 | 0.01 |
|  |  | $L_4$ | 0.07 | 0.02 |
|  |  | $L_5$ | 0.49 | 0.13 |
| T | 0.07 | $T_1$ | 0.56 | 0.13 |
|  |  | $T_2$ | 0.13 | 0.03 |
|  |  | $T_3$ | 0.26 | 0.06 |
|  |  | $T_4$ | 0.05 | 0.01 |
| P | 0.04 | $P_1$ | 0.26 | 0.03 |
|  |  | $P_2$ | 0.47 | 0.06 |
|  |  | $P_3$ | 0.04 | 0.01 |
|  |  | $P_4$ | 0.08 | 0.01 |
|  |  | $P_5$ | 0.14 | 0.02 |
| E | 0.48 | $E_1$ | 0.23 | 0.03 |
|  |  | $E_2$ | 0.65 | 0.07 |
|  |  | $E_3$ | 0.12 | 0.01 |

Therefore, the weight vector **W** of 22 evaluation indexes in the comprehensive evaluation index system can be finally determined.

**W** = (**W**$_{01}$, **W**$_{02}$, **W**$_{03}$, **W**$_{04}$, **W**$_{05}$) = (0.01, 0.13, 0.06, 0.02, 0.03, 0.07, 0.04, 0.01, 0.02, 0.13, 0.13, 0.03, 0.06, 0.01, 0.03, 0.06, 0.01, 0.01, 0.02, 0.03, 0.07, 0.01).

### 3.2. Evaluation Matrix of Evaluation Indexes

FCEM is an objective evaluation method for indexes. This method can change a qualitative subjective judgment into a quantitative objective judgment based on membership theory. The author

collected and analyzed the detailed data of 55 townships in JianYang County and finally determined 22 representative evaluation indexes. According to the reasonable classification, 1210 sets of data were obtained.

The urgency of the human settlement environment (THSE) improvement is affected by many factors. According to the characteristics of THSE in JianYang County, the author divides the urgency of THSE improvement into five grades: grade **A** represents a need to be improved extremely, grade **B** represents a need to be improved urgently, grade **C** represents a need to be improved intensively, grade **D** represents a need to be improved obviously, and grade **E** represents a need to be improved slightly. The five grades are identified by the red, orange, yellow, blue and cyan, respectively. The classification of the human settlement environment (THSE) improvement is shown in Table 7.

**Table 7.** The grading table of the improvement of the human settlement environment (THSE) in JianYang County.

| Improvement Grade | A | B | C | D | E |
|---|---|---|---|---|---|
| Need to be Improved | Extremely | Urgently | Intensively | Obviously | Slightly |
| Color Identification | | | | | |

The author analyzed the characteristics of the evaluation index value in detail and reasonably divided the value range of the evaluation index according to the improvement grade of THSE, as shown in Table 8. Since the larger the value of the Engel coefficient ($F_3$), the higher the grade of improvement, ($50-F_3$) is used as a substitutable index of the Engel coefficient in this table. The interval endpoint values from small to large are defined as $N_1$, $N_2$, $N_3$, $N_4$ in Table 8.

**Table 8.** Value range of the evaluation index and improvement grade.

| Primary Index | Secondary Index | Improvement Grade | | | | |
|---|---|---|---|---|---|---|
| | | A | B | C | D | E |
| F | $F_1$ | ≤2900 | (2900)–4300 | (4300)–5600 | (5600)–6900 | ≥6900 |
| | $F_2$ | ≤1700 | (1700)–2600 | (2600)–3400 | (3400)–4200 | ≥4200 |
| | $50-F_3$ | ≤6.5 | (6.5)–9.5 | (9.5)–12.5 | (12.5)–15.5 | ≥15.5 |
| | $F_4$ | ≤25 | (25)–32 | (32)–40 | (40)–48 | ≥48 |
| | $F_5$ | ≤26 | (26)–32 | (32)–37 | (37)–42 | ≥42 |
| L | $L_1$ | ≤40 | (40)–48 | (48)–60 | (60)–68 | ≥68 |
| | $L_2$ | ≤27 | (27)–34 | (34)–41 | (41)–48 | ≥48 |
| | $L_3$ | ≤43 | (43)–48 | (48)–53 | (53)–58 | ≥58 |
| | $L_4$ | ≤62 | (62)–68 | (68)–74 | (74)–79 | ≥79 |
| | $L_5$ | ≤36 | (36)–40 | (40)–43 | (43)–46 | ≥46 |
| T | $T_1$ | ≤67 | (67)–74 | (74)–80 | (80)–87 | ≥87 |
| | $T_2$ | ≤7 | (7)–13 | (13)–20 | (20)–26 | ≥26 |
| | $T_3$ | ≤50 | (50)–57 | (57)–64 | (64)–70 | ≥70 |
| | $T_4$ | ≤2 | (2)–2.6 | (2.6)–3.2 | (3.2)–3.8 | ≥3.8 |
| P | $P_1$ | ≤1.5 | (1.5)–2.1 | (2.1)–2.7 | (2.7)–3.3 | ≥3.3 |
| | $P_2$ | ≤0.3 | (0.3)–0.4 | (0.4)–0.5 | (0.5)–0.6 | ≥0.6 |
| | $P_3$ | ≤17 | (17)–20 | (20)–23 | (23)–26 | ≥26 |
| | $P_4$ | ≤3.5 | (3.5)–4.5 | (4.5)–5.5 | (5.5)–6.5 | ≥6.5 |
| | $P_5$ | ≤6 | (6)–7 | (7)–9 | (9)–10 | ≥10 |
| E | $E_1$ | ≤56 | (56)–62 | (62)–68 | (68)–74 | ≥74 |
| | $E_2$ | ≤74 | (74)–77 | (77)–80 | (80)–84 | ≥84 |
| | $E_3$ | ≤2.2 | (2.2)–2.9 | (2.9)–3.7 | (3.7)–4.5 | ≥4.5 |

According to the FCEM, it is important to transform the interval value of a single evaluation indel into a certain value by using a membership function. The author constructed a membership function

table for THSE assessment in JianYang County based on the characteristics of THSE and the research results of some other experts and scholars [59,60], as shown in Table 9.

**Table 9.** Membership function table

| Evaluation Interval | Improvement Grade | | | | |
|---|---|---|---|---|---|
| | **A** | **B** | **C** | **D** | **E** |
| $y \leq N_1$ | $1 - \frac{y}{2N_1}$ | $\frac{y}{2N_1}$ | 0 | 0 | 0 |
| $N_1 < y \leq \frac{N_1+N_2}{2}$ | $\frac{(N_1+N_2)-2y}{2(N_2-N_1)}$ | $1 - \frac{(N_1+N_2)-2y}{2(N_2-N_1)}$ | 0 | 0 | 0 |
| $\frac{N_1+N_2}{2} < y \leq N_2$ | 0 | $1 - \frac{2y-(N_1+N_2)}{2(N_2-N_1)}$ | $\frac{2y-(N_1+N_2)}{2(N_2-N_1)}$ | 0 | 0 |
| $N_2 < y \leq \frac{N_2+N_3}{2}$ | 0 | $\frac{(N_2+N_3)-2y}{2(N_3-N_2)}$ | $1 - \frac{2y-(N_2+N_3)}{2(N_3-N_2)}$ | 0 | 0 |
| $\frac{N_2+N_3}{2} < y \leq N_3$ | 0 | 0 | $1 - \frac{2y-(N_2+N_3)}{2(N_3-N_2)}$ | $\frac{2y-(N_2+N_3)}{2(N_3-N_2)}$ | 0 |
| $N_3 < y \leq \frac{N_3+N_4}{2}$ | 0 | 0 | $\frac{(N_3+N_4)-2y}{2(N_4-N_3)}$ | $1 - \frac{(N_3+N_4)-2y}{2(N_4-N_3)}$ | 0 |
| $\frac{N_3+N_4}{2} < y \leq N_4$ | 0 | 0 | 0 | $1 - \frac{2y-(N_3+N_4)}{2(N_4-N_3)}$ | $\frac{2y-(N_3+N_4)}{2(N_4-N_3)}$ |
| $y > N_4$ | 0 | 0 | 0 | $\frac{N_4}{2y}$ | $1 - \frac{N_4}{2y}$ |

In the table, $y$ represents the value of the evaluation index.

This section takes Hongyuan Township (Code Number 1) as an example and uses the FCEM to calculate the evaluation matrix $\mathbf{E_1}$ of the evaluation indexes, $\mathbf{E_1} = (\mathbf{e}_1, \mathbf{e}_2, \dots \mathbf{e}_{22})^T$. The evaluation index values of Hongyuan Township are shown in Table 10.

**Table 10.** Evaluation index value of Hongyuan Township.

| F | $F_1$ | $F_2$ | $F_3$ | $F_4$ | $F_5$ | L | $L_1$ | $L_2$ | $L_3$ | $L_4$ | $L_5$ |
|---|---|---|---|---|---|---|---|---|---|---|---|
| | 3,000.61 | 11,621.76 | 43.7 | 20.1 | 35.4 | | 45.6 | 24.5 | 45.3 | 64.3 | 48.6 |
| T | $T_1$ | $T_2$ | $T_3$ | $T_4$ | | P | $P_1$ | $P_2$ | $P_3$ | $P_4$ | $P_5$ |
| | 73.7 | 0.0 | 50.6 | 2.4 | | | 1.2 | 0.4 | 17.4 | 4.2 | 5.4 |
| E | $E_1$ | $E_2$ | $E_3$ | | | | | | | | |
| | 62.5 | 82.2 | 3.5 | | | | | | | | |

When the evaluation index is per capita local GDP ($F_1$), according to Tables 8 and 9:
$N_1 = 2{,}900$; $N_2 = 4{,}300$; $N_3 = 5{,}600$; $N_4 = 6{,}90$; $\frac{N_1+N_2}{2} = 3{,}600$; $\frac{N_2+N_3}{2} = 4{,}950$; $\frac{N_3+N_4}{2} = 6{,}250$
$y = 3{,}000.61$
Obviously: $N_1(2,900) < y(3,000.61) \leq \frac{N_1+N_2}{2}(3,600)$
$\mathbf{e}_1 = (\frac{(N_1+N_2)-2y}{2(N_2-N_1)}, 1 - \frac{(N_1+N_2)-2y}{2(N_2-N_1)}, 0, 0, 0)$
$\frac{(N_1+N_2)-2y}{2(N_2-N_1)} = 0.43$, $1 - \frac{(N_1+N_2)-2y}{2(N_2-N_1)} = 0.57$
Therefore, $\mathbf{e}_1 = (0.43, 0.57, 0, 0, 0)$.

In the same way, the evaluation vector $\mathbf{e}_i$ of other evaluation indexes can be calculated. Therefore, $\mathbf{E_1} = (\mathbf{e}_1, \mathbf{e}_2, \dots \mathbf{e}_{22})^T$, that is,

$$\mathbf{E_1} = \begin{pmatrix} 0.43 & 0.52 & 0.52 & 0.60 & 0.00 & 0.00 & 0.55 & 0.04 & 0.12 & 0.00 & 0.64 & 1.00 & 0.41 & 0.00 & 0.60 & 0.00 & 0.37 & 0.00 & 0.55 & 0.00 & 0.00 & 0.00 \\ 0.57 & 0.48 & 0.48 & 0.40 & 0.00 & 0.80 & 0.45 & 0.96 & 0.88 & 0.00 & 0.36 & 0.00 & 0.59 & 0.83 & 0.40 & 0.50 & 0.63 & 0.80 & 0.45 & 0.42 & 0.00 & 0.00 \\ 0.00 & 0.00 & 0.00 & 0.00 & 0.82 & 0.20 & 0.00 & 0.00 & 0.00 & 0.00 & 0.00 & 0.00 & 0.00 & 0.17 & 0.00 & 0.50 & 0.00 & 0.20 & 0.00 & 0.58 & 0.00 & 0.75 \\ 0.00 & 0.00 & 0.00 & 0.00 & 0.18 & 0.00 & 0.00 & 0.00 & 0.00 & 0.47 & 0.00 & 0.00 & 0.00 & 0.00 & 0.00 & 0.00 & 0.00 & 0.00 & 0.00 & 0.00 & 0.95 & 0.25 \\ 0.00 & 0.00 & 0.00 & 0.00 & 0.00 & 0.00 & 0.00 & 0.00 & 0.00 & 0.53 & 0.00 & 0.00 & 0.00 & 0.00 & 0.00 & 0.00 & 0.00 & 0.00 & 0.00 & 0.00 & 0.05 & 0.00 \end{pmatrix}^T$$

### 3.3. Comprehensive Evaluation Results

The weight vector **W** and the evaluation matrix **E** have been calculated in Sections 3.1 and 3.2 of this paper, respectively. In this section, Hongyuan Township is still used as an example to solve the comprehensive evaluation matrix **C**. According to the theory of fuzzy mathematics, the fuzzy decision-making process of the multi-objective evaluation system can be represented by the comprehensive evaluation matrix **C**, $\mathbf{C} = \mathbf{W} \bullet \mathbf{E} = (c_1, c_2, \dots c_g)$, and the maximum value of $c_g$ corresponds

to the evaluation grade of the research object. Therefore, the comprehensive evaluation matrix $\mathbf{C}_1$ of Hongyuan Township can be calculated by matrix operation:

$\mathbf{C}_1 = \mathbf{W} \bullet \mathbf{E}_1 = (0.01, 0.13, 0.06, 0.02, 0.03, 0.07, 0.04, 0.01, 0.02, 0.13, 0.13, 0.03, 0.06, 0.01, 0.03, 0.06, 0.01, 0.01, 0.02, 0.03, 0.07, 0.01) \bullet$

$$
\begin{pmatrix}
0.43 & 0.52 & 0.52 & 0.60 & 0.00 & 0.00 & 0.55 & 0.04 & 0.12 & 0.00 & 0.64 & 1.00 & 0.41 & 0.00 & 0.60 & 0.00 & 0.37 & 0.00 & 0.55 & 0.00 & 0.00 & 0.00 \\
0.57 & 0.48 & 0.48 & 0.40 & 0.00 & 0.80 & 0.45 & 0.96 & 0.88 & 0.00 & 0.36 & 0.00 & 0.59 & 0.83 & 0.40 & 0.50 & 0.63 & 0.80 & 0.45 & 0.42 & 0.00 & 0.00 \\
0.00 & 0.00 & 0.00 & 0.00 & 0.82 & 0.20 & 0.00 & 0.00 & 0.00 & 0.00 & 0.00 & 0.00 & 0.00 & 0.17 & 0.00 & 0.50 & 0.00 & 0.20 & 0.00 & 0.58 & 0.00 & 0.75 \\
0.00 & 0.00 & 0.00 & 0.00 & 0.18 & 0.00 & 0.00 & 0.00 & 0.00 & 0.47 & 0.00 & 0.00 & 0.00 & 0.00 & 0.00 & 0.00 & 0.00 & 0.00 & 0.00 & 0.00 & 0.95 & 0.25 \\
0.00 & 0.00 & 0.00 & 0.00 & 0.00 & 0.00 & 0.00 & 0.00 & 0.00 & 0.53 & 0.00 & 0.00 & 0.00 & 0.00 & 0.00 & 0.00 & 0.00 & 0.00 & 0.00 & 0.00 & 0.05 & 0.00
\end{pmatrix}^T
=
\begin{pmatrix}
0.31 \\
0.38 \\
0.10 \\
0.14 \\
0.07
\end{pmatrix}^T
$$

The five comprehensive evaluation values of $\mathbf{C}_1$ correspond to the five improvement grades, respectively, as shown in Table 11.

**Table 11.** Corresponding table of comprehensive evaluation values and improvement grades.

| Improvement Grade | A | B | C | D | E |
|---|---|---|---|---|---|
| $\mathbf{C}_g$ ($g$=1, 2, … 5) | 0.31 | **0.38** | 0.10 | 0.14 | 0.07 |

Therefore, the improvement grade of the human settlement environment (THSE) in Hongyuan Township is grade **B**: i.e., it needs to be improved urgently.

The improvement grade of THSE of the other 54 townships is similar to the calculation process of Hongyuan Township. During the calculation process in this article, more than 30,000 data operations are involved. In order to express the final research results clearly and concisely, the author lists the final calculation results as follows, as shown in Table 12.

In order to better display the different improvement grade of the 55 townships, the author drew an improvement grade map of the human settlement environment (THSE) by using the map and color annotation method in JianYang County, as shown in Figure 3. The number in the figure represents the township code number, and the circled background color represents the improvement grade.

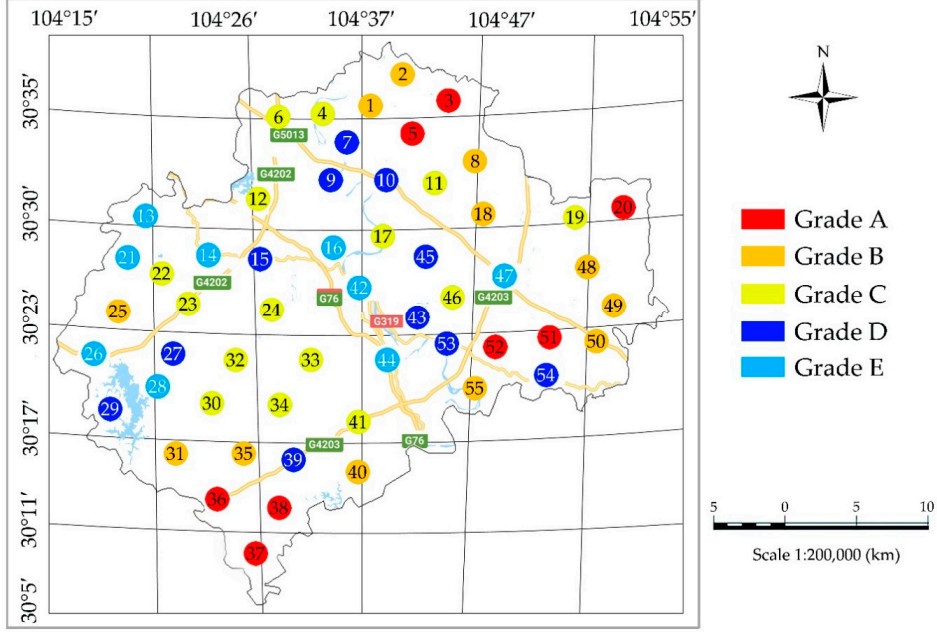

**Figure 3.** The improvement grade map of THSE in JianYang County.

**Table 12.** Table of improvement grade of the human settlement environment (THSE) in JianYang County.

| Code Number | $C= W\bullet E= (c_1, c_2, c_3, c_4, c_5)$ | | | | | Grade | Code Number | $C= W\bullet E= (c_1, c_2, c_3, c_4, c_5)$ | | | | | Grade | Code Number | $C= W\bullet E= (c_1, c_2, c_3, c_4, c_5)$ | | | | | Grade |
|---|---|---|---|---|---|---|---|---|---|---|---|---|---|---|---|---|---|---|---|---|
| 1 | 0.31 | **0.38** | 0.10 | 0.14 | 0.07 | B | 20 | **0.43** | 0.22 | 0.18 | 0.13 | 0.04 | A | 39 | 0.25 | 0.15 | 0.23 | **0.36** | 0.01 | D |
| 2 | 0.27 | **0.29** | 0.18 | 0.16 | 0.10 | B | 21 | 0.23 | 0.07 | 0.23 | 0.14 | **0.33** | E | 40 | 0.27 | 0.34 | 0.16 | 0.17 | 0.06 | B |
| 3 | **0.33** | 0.26 | 0.11 | 0.20 | 0.10 | A | 22 | 0.16 | 0.33 | **0.34** | 0.14 | 0.03 | C | 41 | 0.35 | 0.09 | **0.38** | 0.12 | 0.06 | C |
| 4 | 0.21 | 0.13 | **0.34** | 0.21 | 0.11 | C | 23 | 0.12 | 0.18 | **0.38** | 0.27 | 0.05 | C | 42 | 0.24 | 0.23 | 0.05 | 0.19 | **0.29** | E |
| 5 | **0.32** | 0.26 | 0.19 | 0.17 | 0.05 | A | 24 | 0.14 | 0.28 | **0.33** | 0.20 | 0.05 | C | 43 | 0.19 | 0.23 | 0.20 | 0.25 | 0.13 | D |
| 6 | 0.24 | 0.32 | **0.36** | 0.06 | 0.02 | C | 25 | 0.12 | **0.36** | 0.32 | 0.18 | 0.03 | B | 44 | 0.20 | 0.09 | 0.20 | 0.23 | **0.28** | E |
| 7 | 0.16 | 0.28 | 0.15 | **0.39** | 0.01 | D | 26 | 0.15 | 0.10 | 0.15 | 0.26 | **0.34** | E | 45 | 0.24 | 0.21 | 0.20 | **0.29** | 0.06 | D |
| 8 | 0.32 | **0.38** | 0.20 | 0.07 | 0.02 | B | 27 | 0.18 | 0.18 | 0.27 | **0.32** | 0.05 | D | 46 | 0.23 | 0.19 | **0.28** | 0.26 | 0.04 | C |
| 9 | 0.16 | 0.15 | 0.21 | **0.39** | 0.09 | D | 28 | 0.23 | 0.22 | 0.08 | 0.10 | **0.37** | E | 47 | 0.26 | 0.03 | 0.19 | 0.22 | **0.29** | E |
| 10 | 0.27 | 0.21 | 0.10 | **0.36** | 0.05 | D | 29 | 0.14 | 0.17 | 0.21 | **0.41** | 0.07 | D | 48 | 0.29 | **0.33** | 0.21 | 0.16 | 0.02 | B |
| 11 | 0.13 | 0.30 | **0.34** | 0.13 | 0.10 | C | 30 | 0.16 | 0.24 | **0.29** | 0.24 | 0.07 | C | 49 | 0.25 | **0.28** | 0.11 | 0.24 | 0.13 | B |
| 12 | 0.32 | 0.11 | **0.49** | 0.08 | 0.00 | C | 31 | 0.15 | **0.38** | 0.24 | 0.15 | 0.08 | B | 50 | 0.28 | **0.33** | 0.17 | 0.18 | 0.05 | B |
| 13 | 0.13 | 0.21 | 0.16 | 0.06 | **0.44** | E | 32 | 0.22 | 0.24 | **0.47** | 0.05 | 0.02 | C | 51 | **0.39** | 0.28 | 0.02 | 0.16 | 0.15 | A |
| 14 | 0.20 | 0.08 | 0.22 | 0.12 | **0.39** | E | 33 | 0.13 | 0.29 | **0.37** | 0.16 | 0.05 | C | 52 | **0.35** | 0.20 | 0.12 | 0.27 | 0.06 | A |
| 15 | 0.25 | 0.10 | 0.28 | **0.35** | 0.03 | D | 34 | 0.19 | 0.29 | **0.38** | 0.13 | 0.00 | C | 53 | 0.28 | 0.28 | 0.12 | **0.29** | 0.03 | D |
| 16 | 0.15 | 0.24 | 0.25 | 0.06 | **0.30** | E | 35 | 0.31 | **0.28** | 0.22 | 0.16 | 0.03 | B | 54 | 0.12 | 0.30 | 0.20 | **0.29** | 0.09 | D |
| 17 | 0.17 | 0.16 | **0.34** | 0.30 | 0.03 | C | 36 | **0.35** | 0.15 | 0.18 | 0.20 | 0.12 | A | 55 | 0.24 | **0.28** | 0.10 | 0.25 | 0.13 | B |
| 18 | 0.27 | **0.30** | 0.29 | 0.13 | 0.01 | B | 37 | **0.26** | 0.24 | 0.24 | 0.15 | 0.12 | A | | | | | | | |
| 19 | 0.16 | 0.23 | **0.33** | 0.21 | 0.07 | C | 38 | **0.32** | 0.26 | 0.25 | 0.14 | 0.03 | A | | | | | | | |

In order to further determine the order of improvement of the human settlement environment (THSE) in the 55 townships, the author made a horizontal comparison of townships under the same improvement grade. According to the theory of fuzzy mathematics, the larger the value of the factor $c_g$ in the comprehensive evaluation matrix **C**, the more urgently the research object needs to be improved. The horizontal comparison diagram is shown in Figure 4.

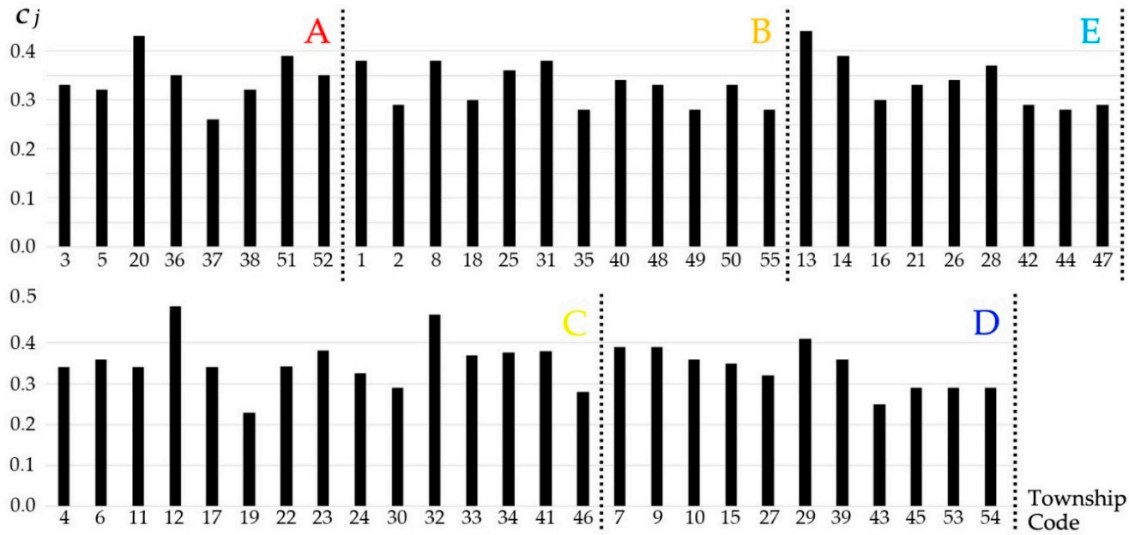

**Figure 4.** Horizontal comparison diagram under the same improvement grade.

## 4. Discussion and Conclusions

According to the research results in this article, the Chengdu Municipal Government can scientifically and reasonably determine the improvement order of the human settlement environment (THSE) of 55 townships in JianYang County with reference to Table 12, Figures 3 and 4. The assessment method of the human settlement environment (THSE) in this paper can overcome the blindness of decision-makers in making decisions based only on subjective experience or a one-sided index.

Previous studies on the human settlement environment (THSE) mainly focused on qualitative or semi qualitative studies, with relatively few quantitative studies reported. Quantitative research has the advantages of clear presentation, scientific analysis process and high reliability of the results, but there are few suitable quantitative evaluation methods for the assessment of THSE. The author has been engaged in the research of THSE for many years and has investigated the living environment of more than 300 townships in Western China. After trialing dozens of analysis methods, the author found that the FAHP is a very suitable quantitative analysis method for THSE assessment.

In this paper, the author used FAHP to study the improvement grade of THSE. Compared with qualitative or semi-quantitative methods, FAHP has obvious advantages in both theory and data processing. The method of the FAHP has a perfect theoretical system, a simple calculation process, and an evaluation process consistent with human thinking. Moreover, in the process of solving practical problems, this method not only fully considers the expert's experience and knowledge but also has a rigorous mathematical formula to support it, which is very suitable for solving the problem of human settlement environment assessment.

In the process of the human settlement environment (THSE) assessment, two basic methods of the FAHP were adopted: rigorous theoretical derivation and expert evaluation and scoring to deal with the research objects and evaluation indexes. The selection process of evaluation indexes, expert evaluation processes, and scoring processes are always dynamic. The author will choose the appropriate evaluation indexes according to the actual situation between the research objects, and the experts will make dynamic evaluations according to the difference of the research objects and the

indicators. That is to say, using the FAHP to evaluate THSE has the integrity of the theoretical system and the real-time effectiveness of dynamic evaluation.

The problem-solving process of the FAHP can be divided into four basic calculation steps. In order to ensure the integrity of the calculation structure, the key calculation steps of the FAHP are preserved in the paper. The calculation process has also been streamlined to the maximum extent possible. In order to ensure the readability of the paper and repeatability of the research process, the author selected representative data for example calculations in each calculation process. The final calculation results are summarized in Tables 4, 6 and 10–12, respectively.

A simple and easy-to-use online assessment tool is an important means of preliminary understanding of the object. However, more rigorous and scientific calculation methods must be used for in-depth research. The FAHP used in this paper is a core algorithm that can solve complex social science problems. The author applied the FAHP to the assessment and improvement of THSE in JianYang county and reasonably evaluated the improvement grade of 55 townships based on a large amount of survey data and rigorous theoretical derivation. The Urban Benchmarking Tool and the CityBench Tool developed by the ESPON 2013 Programme are easy-to-use "quick-scan" web tools, which are mainly used to quickly evaluate the development prospects and investment potential of European cities. Although the tools cannot be directly applied to this study, its visual online evaluation method can provide useful references for the author's follow-up research. With more and more data to be obtained by the author's team in the follow-up work, the development of a visual assessment tool for the human settlement environment in Chengdu city is also an important research direction for us in the future.

After the author's team has submitted the preliminary investigation report of the human settlement environment (THSE) in JianYang County, it will start to carry out the follow-up research on how to specifically realize the improvement of THSE. Generally speaking, the overall requirements of a "more beautiful ecological environment, more comfortable living condition, more prosperous social culture, more convenient transportation, and significantly higher income level" should be implemented. As the FAHP has universal practicability, the research results of this paper are also applicable to the assessment and improvement of the human settlement environment (THSE) in other areas. Taking Chengdu as an example, as a city with a history of thousands of years, there are a large number of old areas in the main urban area. In the process of building a central city in Western China, the FAHP can be used to make a reasonable assessment and improvement of these old human settlements. Obviously, the situation in Chengdu city is much more complicated than that in JianYang County. Therefore, how to solve the problem of THSE in areas with more indexes and more complex data will be the main research direction of the author in the future.

The FAHP also has some limitations, such as: (1) When there are too many evaluation indexes, the corresponding statistical data will be greater. Since this method involves matrix operation, the calculation workload in the analysis process will be larger. (2) The calculation process of the eigenvector and the largest eigenvalue in the FAHP is complicated. (3) When scoring by experts, if the number of experts is insufficient or the evaluation is not reasonable, there will be some calculation errors. Therefore, in the specific application process, the FAHP should be used appropriately according to the situation.

**Supplementary Materials:** The following are available online at http://www.mdpi.com/2071-1050/12/4/1563/s1, File S1: Chengdu's overall plan for implementing the strategy of 'Eastward' (2018–2035). Figure S1: The location of the study area.; Figure S2: The geographical location map (3D Space Diagram); Figure S3: The improvement grade map of THSE in JianYang County; Figure S4: Horizontal comparison diagram under the same improvement grade.

**Author Contributions:** Conceptualization, Y.Z. and Q.F.; methodology, Y.Z. and Q.F.; validation, Y.Z. and Q.F.; formal analysis, Y.Z.; investigation, Y.Z. and Q.F.; data curation, Y.Z. and Q.F.; writing—original draft preparation, Y.Z. and Q.F.; writing—review and editing, Y.Z.; supervision, Y.Z. and Q.F. All authors have read and agreed to the published version of the manuscript.

**Funding:** This study was supported by the National Key Research and Development Program of China (No. 2016YFC0401603); National Natural Science Foundation of China (51879178); National Social Science Foundation in China (No. 15BKS038).

**Conflicts of Interest:** The authors declare no conflict of interest.

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
