# Peer review of "The Application of the Fuzzy Analytic Hierarchy Process in the Assessment and Improvement of the Human Settlement Environment"

_sustainability, doi:10.3390/su12041563_

Round 1

Reviewer 1 Report

The article is quite interesting. It is an application of the Fuzzy Analytic Hierarchy Process for supporting decision-making in assessing and improving the human settlement environment in a county of China, using a rather old      ( Ekistics theory etc.) and static theoretical background. This means that linking with the development process of human settlements should be more seriously envisaged. 

The calculation part of the article is too extensive and complete. The comparison with other methodologies is quite extensive and should be limited.

There are repetitions that should be avoided ( please see attached document, highlights in green) 

The research is in fact an urban benchmarking process, adapted to the situation of urban settlements in China. There is a great part comparing the FAHP method to other similar tools but what should be needed on my opinion is rather the comparison of the benchmarking/decision-making tool constructed by the authors with the Urban Benchmarking Tool and the CityBench Tool development in the ESPON programme. This would add substantially to the scientific soundness of the research. 

I think that the article should be improved and "modernised", seriously taking into account the tool of Urban Benchmarking as developed in Europe and especially through the ESPON Programme 2007-2013 (www.espon.eu)  

The essence of urban benchmarking is the comparison of indicators describing a given territorial unit, e.g., a city or metropolitan area, with similar indicators describing other units. As a result, there is a clear diagnosis of the level of development of a unit as compared to a selected reference group.Urban benchmarking also serves to analyse and evaluate peculiarities, trends and challenges of cities. It is executed based on qualitative (e.g. quality of life) and quantitative data assessed with regard to a special issue. Such issues cover social, economic and ecological topics, but often also a combination of them in order to gain better insight. Benchmarking enables a city to generate information allowing for comparison with other cities and regions in order to identify its own position within the national or even international urban system and, if necessary, to develop strategies for action. Owing to the competition of cities concerning investors, well-educated population sections and institutions, benchmarking is increasingly gaining importance, which is reflected by the growing number of local activities in this area.

I would advise the authors to have a look on the following sources and links : 

1. https://www.espon.eu/tools-maps/citybench-urban-benchmarking

2. EC (2013), Quality of life in cities Perception survey in 79 European cities Flash Eurobarometer 366, Luxembourg: Publications Office of the European Union, 2013

3. CityBenCh Web tool:

http://www.geotec.uji.es/european-urban-benchmarking-webtool/

4. http://www.europeancitiesmarketing.com/research/reports-and-studies/city- tourism-benchmark-tool/

5. http://www.urbanaudit.org/ http://www.eea.europa.eu/data-and-maps/data/urban-atlas

I hope these can be of help for the authors to improve their paper. 

Author Response

Dear Professor:

       The point-by-point response to your kind comments and suggestions is contained in the PDF file, named "Response to reviewer 1". Please see the attachment.

Reviewer 2 Report

The topic of this paper seems to be interesting. The objectives of the study are clear. However, the authors should revise their manuscript by following my comments below:

- In Discussion and Conclusion, Table 12 and Figures 3 and 4 should be better placed in the section 3: The Application of FAHP and Results.

- Also, In Section 4, the authors need to integrate their findings better with those of previous studies.

- Finally, the authors should add some limitations of their study.

Good luck with the revision

Author Response

Dear Professor:

       The point-by-point response to your kind comments and suggestions is contained in the PDF file, named "Response to reviewer 2". Please see the attachment.

Round 2

Reviewer 1 Report

The authors have made substantial effort to improve their manuscript and respond to the Reviewer comments. 

Although I still have some theoretical reservations, I sincerely think that the research is complete according to the authors' research questions and expectations and the manuscript can be accepted for publication in current form. Please check all the DOIs in the references there are some of them that are not opening. 

Author Response

Dear Professor:

We are very grateful for your recognition of our paper. The constructive suggestions you put forward in the review process are of great significance and value to the improvement of the quality of our paper. Some minor errors mentioned by reviewer have been modified. What's more, the author has carried on a careful examination to the paper, and some minor grammatical errors were corrected. Revised words are marked in red in the manuscript and "Track Changes" function was used in Microsoft Word.

We hope the revised content can get your approval. Once again, thank you very much for your comments and suggestions. I wish you every success. Have a nice day!

Yours sincerely,

The author

Reviewer 2 Report

The paper has been significantly improved. I can only indicate this error:

- Page 18, line 425, “Discussion and conclusion” section is duplicated.

Author Response

(The authors gave the same response as above.)
